

# Tackling the phylogenetic conundrum of Hydroidolina (Cnidaria: Medusozoa: Hydrozoa) by assessing competing tree topologies with targeted high-throughput sequencing

Bastian Bentlage[1] and Allen G. Collins[2]

[1] Marine Laboratory, University of Guam, Mangilao, Guam, USA
[2] National Museum of Natural History & National Systematics Laboratory of NOAA's Fisheries Service, Smithsonian Institution, Washington, DC, USA

## ABSTRACT

Higher-level relationships of the Hydrozoan subclass Hydroidolina, which encompasses the vast majority of medusozoan cnidarian species diversity, have been elusive to confidently infer. The most widely adopted phylogenetic framework for Hydroidolina based on ribosomal RNA data received low support for several higher level relationships. To address this issue, we developed a set of RNA baits to target more than a hundred loci from the genomes of a broad taxonomic sample of Hydroidolina for high-throughput sequencing. Using these data, we inferred the relationships of Hydroidolina using maximum likelihood and Bayesian approaches. Both inference methods yielded well-supported phylogenetic hypotheses that largely agree with each other. Using maximum likelihood and Baysian hypothesis testing frameworks, we found that several alternate topological hypotheses proposed previously may be rejected in light of the genomic data generated for this study. Both the maximum likelihood and Bayesian topologies inferred herein consistently score well across testing frameworks, suggesting that their consensus represents the most likely phylogenetic hypothesis of Hydroidolina. This phylogenetic framework places Aplanulata as sister lineage to the remainder of Hydroidolina. This is a strong deviation from previous phylogenetic analyses that placed Capitata or Siphonophorae as sister group to the remainder of Hydroidolina. Considering that Aplanulata represents a lineage comprised of species that for the most part possess a life cycle involving a solitary polyp and free-swimming medusa stage, the phylogenetic hypotheses presented herein have potentially large implications for clarifying the evolution of life cycles, coloniality, and the division of labor in Hydrozoa as taxon sampling for phylogenetic analyses becomes more complete.

## INTRODUCTION

While the fossil record of medusozoan cnidarians is scant, the origin of the group has been inferred to lie near the end of the Ediacaran, approximately 550–580 million years ago

Corresponding author
Bastian Bentlage,
bentlageb@triton.uog.edu

(*Han, Zang & Komiya, 2016*). Plausible crown-group hydrozoans have been described from some 500 million year old Cambrian deposits (*Cartwright et al., 2007*), suggesting an ancient origin of extant hydrozoans likely dating back to the period of rapid diversification of metazoan life during which all major modern animal phyla emerged (*Valentine, Jablonski & Erwin, 1999*; *Erwin, 2020*). Hydrozoans are of particular interest in the study of the evolution of development, as their radiation gave rise to diverse life cycle strategies, diverse forms of coloniality and the division of labor (*Cartwright & Nawrocki, 2010*; *Bentlage et al., 2018*; *Cartwright, Travert & Sanders, 2020*). This diversity is concentrated in the hydrozoan subclass Hydroidolina, the medusozoan clade that contains the vast majority of the 3,800 nominal hydrozoan species (*Daly et al., 2007*; *Schuchert, 2020*). Elucidating the evolutionary history and patterns of complex character evolution in Hydroidolina requires a solid understanding of the phylogenetic history of the group (*e.g.*, *Cartwright, Travert & Sanders, 2020*).

However, the goal of inferring the deep phylogeny of Hydroidolina has been elusive, possibly as a result of the early origin and likely rapid diversification of the group. The most comprehensive phylogenetic hypothesis (*Cartwright & Nawrocki, 2010*) of higher-level relationships within Hydroidolina was inferred using ribosomal DNA (rDNA) from a broad taxonomic sample. While shallow nodes were well resolved with high confidence, higher-level relationships generally received weak support. In particular, Leptothecata, Siphonophorae, Capitata, and Aplanulata were inferred to be monophyletic groups while Filifera was polyphyletic (*Cartwright et al., 2008*; *Cartwright & Nawrocki, 2010*). Previously, the latter three taxa were united under the name Anthoathecata but their non-monophyly had been demonstrated earlier (*Collins et al., 2006*). Whole mitochondrial genomes have been employed previously to address the issue of reconstructing deep nodes within the phylogeny of Hydroidolina (*Kayal et al., 2015*). While this approach led to a well-resolved and highly supported phylogenetic hypothesis, several nodes of the resulting tree topology are at odds with rDNA-based phylogenies (*Cartwright et al., 2008*; *Cartwright & Nawrocki, 2010*) and recent phylogenomics-based hypotheses (*Kayal et al., 2018*).

Advances in understanding of medusozoan, and more broadly cnidarian relationships, were made by employing phylogenomic datasets derived from whole-transcriptome and genome-sequencing efforts (*Zapata et al., 2015*; *Kayal et al., 2018*). While these efforts provided answers to several long-standing questions of cnidarian evolutionary history, taxon sampling was insufficient for rigorous evaluation of hydrozoan relationships. We used the coding sequences generated by these phylogenomic studies as a backbone for targeted high-throughput sequencing, producing a multi-locus dataset to infer the phylogeny of Hydroidolina. For this purpose, we developed a set of custom baits to enrich target loci from a representative sample of hydroidolinan hydrozoans. This work provides a new framework for the phylogeny of Hydroidolina that will enable further phylogenetic comparative studies of character evolution in Hydrozoa. To evaluate multiple competing topological hypotheses of hydroidolinan relationships, both likelihood and Bayesian statistical frameworks (cf. *Sober, 2008*) were employed to discriminate between alternate tree topologies and evaluate the strength of evidence supporting prior phylogenetic

hypotheses of Hydroidolina as well as those phylogenies inferred from the multi-locus dataset presented here.

## MATERIALS AND METHODS

### Bait design for targeted sequencing

Biotinylated RNA baits of 120 bp length were designed using 355 orthogroup partitions (loci) from the phylogenomic OF-PTP_75tx matrix (*Kayal et al., 2018*). This dataset contains the coding sequences from a broad taxonomic sample of cnidarians, including anthozoans, medusozoans, and endocnidozoans. Due to their large sequence divergences, endocnidozoans were excluded from the dataset used for RNA bait design. Nucleotide sequences were aligned per locus guided by their predicted amino acid translations using the Geneious aligner (version 9; Biomatters, Auckland, New Zealand). Several sequences were flagged after manual audit of alignments due to their apparent divergence and identified as non-homologous sequences after BLAST searches against NCBI's GenBank. Following curation and removal of sequences shorter than 80 bp, putative repetitive elements in the remaining 5,523 sequences were soft-masked using the cross_match search option against the Cnidaria repeats database in RepeatMasker (version 4.06; *Smit, Hubley & Green, 2013-2015*).

Next, sequence stretches of 10 or fewer ambiguous nucleotides (N) were replaced with thymine (T) repeats to allow bait design across short regions of undetermined sequences. Similarly, short sequences less than 120 bp were padded with T repeats to allow for 120 bp long baits to be mapped against reference sequences. Baits were designed at approximately 2.5× tiling density with some 50 bp spacing between the start of neighboring baits, yielding 38,102 candidate baits. Each candidate bait was verified against six cnidarian genome assemblies (Table 1) using BLAST searches to evaluate the bait's fit to the reference. Hits with a length greater than 45 bp in length and identity greater than 75% were retained for further analysis. Melting temperatures (Tm; defined as the temperature at which 50% of molecules hybridize) were estimated for each BLAST hit assuming standard Mybaits (Mycroarray, Arbor Biosciences, Ann Arbor, MI, USA) hybridization conditions. Based on the distribution of inferred Tms, baits of moderate or higher specificity were retained. That is, candidate baits with at most 10 BLAST hits between 62.5 °C to 65 °C, two BLAST hits above 65 °C, and fewer than 2 baits on each flanking region were retained, yielding a total of 37,546 baits (Supplemental File 1).

### Target capture and sequencing

Genomic DNA was extracted from ethanol-preserved tissue samples (Table 2) using a standard organic phenol-chloroform extraction protocol (*Green & Sambrook, 2012*). Extracted DNA was quantified with a Qubit 4 fluorometer (Thermo Fisher Scientific, Waltham, MA, USA). 50 μl of DNA in TE buffer were chilled to 4 °C in the water bath of a Q800 Sonicator (Qsonica, Newton, CT, USA) and acoustically sheared for nine minutes using an amplitude of 25 with sonication pulses of 15s on/15s off. Illumina sequencing libraries were constructed from sheared DNA samples using an NEBNext Ultra II DNA library preparation kit (New England Biolabs, Ipswich, MA, USA) with dual indexes

**Table 1 Reference genomes used for verification of candidate baits.** Genome assemblies were obtained from the National Institute of Health's (NCBI) Genbank database.

| Taxon | BioProject Accession | Reference |
|-------|---------------------|-----------|
| Medusozoa: Hydrozoa: *Hydra magnipapillata* | PRJNA12876 | *Chapman et al. (2010)* |
| Medusozoa: Cubozoa: *Alatina alata* | PRJNA312373 | *Ohdera et al. (2019)* |
| Medusozoa: Staurozoa: *Calvadosia cruxmelitensis* | PRJEB23739 | *Ohdera et al. (2019)* |
| Anthozoa: Pentalucea: *Renilla reniformis* | PRJEB20462 | *Ohdera et al. (2019)* |
| Anthozoa: Scleractinia: *Acropora digitifera* | PRJDA67425 | *Shinzato et al. (2011)* |
| Anthozoa: Actiniaria: *Nematostella vectensis* | PRJNA12581 | *Putnam et al. (2007)* |

for multiplexing following the manufacturer's protocol. After library amplification and magnetic bead purification using Ampure beads (Promega, Madison, WI, USA), amplicons longer or shorter than approximately 200 bp were removed using a BluePippin size select gel eletrophoresis system (Sage Science, Beverly, MA, USA). Concentrations of size-selected libraries were equilibrated, followed by pooling of libraries three to four samples deep for target enrichment.

Hybridization of RNA baits to pooled libraries followed the Mybaits version 3.02 protocol (Mycroarray, Arbor Biosciences, Ann Arbor, MI, USA) with the following modifications. After initial denaturation and blocking with Illumina adapter-specific oligonucleotides, RNA baits were allowed to hybridize for 19 h at 65 °C, 19 h at 60 °C, and 10 h at 55 °C. We used this touchdown procedure on the newly developed and untested bait set in an effort to increase on-target specificity while allowing for sensitivity of reactions. Following hybridization, biotinylated baits were bound to streptavidin-coated magnetic beads (Dynabeads MyOne Streptavidin; Thermo Fisher Scientific, Waltham, MA, USA), followed by stringent washing to remove unbound DNA library molecules. Captured libraries were amplified while bound to beads using KAPA HiFi DNA polymerase and HotStart ReadyMix (Roche, Basel, Switzerland) following the manufacturers protocol. The annealing temperature during the 15 amplification cycles was 60 °C. PCR reactions were cleaned using Ampure magentic beads (Thermo Fisher Scientific, Waltham, MA, USA), washing twice with 80% ethanol. DNA concentrations were quantified fluorometrically, followed by equilibration of target-enrichment pools to equimolar concentrations. All reactions were pooled and the size range of the pool selected for an average length of 450 bp using the BluePippin size select system (Sage Science, Beverly, MA, USA). Following quantification using qPCR, 300bp paired-end reads were generated on the Illumina MiSeq platform with the v3 reagent kit (Illumina, San Diego, CA, USA).

## Sequence assembly, alignment, and gene tree discordance

Sequencing adapters were removed from paired-end sequencing reads using Trimmomatic (version 0.22; *Bolger, Lohse & Usadel, 2014*). Reads were trimmed further using a sliding window of size four, with an average quality of 15 or greater required for bases within the window to be retained. Quality trimmed reads shorter than 75 bp were discarded. The HybPiper pipeline (version 1.2; *Johnson et al., 2016*) was used to identify target sequences from enriched sequencing libraries by comparing all quality trimmed reads against the

**Table 2 Species and sequence data sampled for phylogenetic analyses.** Data for species in bold were generated in this study. For each species, the number of loci and amino acid residues included in the final concatenated 134 locus alignment are provided. NCBI: National Center for Biotechnology Information; USNM: National Museum of Natural History, Smithsonian Institution; UOGCVC: University of Guam Coral Voucher Collection.

| | Taxon | Voucher/Reference | Loci | Residues | NCBI Accession |
|---|---|---|---|---|---|
| | | Hydroidolina | | | |
| Capitata | *Zancleopsis tentaculata* | USNM1622168 | 12 | 1,103 | MW272249–60 |
| | *Millepora dichotoma* | UOGCVC947 | 6 | 726 | MW272119– 24 |
| | *Pennaria disticha* | USNM1622068 | 47 | 4,403 | MW272261–307 |
| Filifera I | *Myrionema hargitti* | USNM1622176 | 22 | 2,005 | MW272227–48 |
| Filifera II | *Proboscidactyla* sp. | USNM1622170 | 5 | 448 | MW272435–39 |
| Leptothecata | *Dynanema crisioides* | USNM1622069 | 62 | 6,173 | MW272032–93 |
| | *Nemalecium lighti* | USNM1622067 | 54 | 5,256 | MW272125–78 |
| | *Aglaophenia parvula* | USNM1621045 | 39 | 3,890 | MW272179–217 |
| | *Kirchenpaueria halecioides* | USNM1622065 | 25 | 2,548 | MW272094–118 |
| | *Kirchenpaueria* sp. | USNM1621044 | 84 | 8,018 | MW272308–91 |
| | *Octophialucium* sp. | USNM1622151 | 40 | 3,334 | MW271992–031 |
| | *Clytia hemispherica* | *Kayal et al. (2018)* | 26 | 4,266 | PRJEB32541 |
| Filifera III | *Podocoryne carnea* | *Kayal et al. (2018)* | 109 | 16,709 | SRX529566 |
| | *Podocoryne martinicana* | USNM1622132 | 9 | 1,077 | MW272218–26 |
| | *Hydractinia symbiolongicarpus* | *Kayal et al. (2018)* | 104 | 16,411 | SRX474878 |
| | *Hydractinia polyclina* | *Kayal et al. (2018)* | 107 | 16,950 | SRR923509 |
| Filifera IV | *Merga violacea* | USNM1622162 | 10 | 831 | MW271982–91 |
| Siphonophorae | *Athorybia rosacea* | USNM1622138 | 6 | 871 | MW272429–34 |
| | *Agalma elegans* | *Kayal et al. (2018)* | 99 | 15,170 | SRX288285 |
| | *Nanomia bijuga* | *Kayal et al. (2018)* | 96 | 14,339 | SRX288430 |
| | *Craseoa lathetica* | *Kayal et al. (2018)* | 89 | 13,177 | SRX288432 |
| | *Abylobsis tetragona* | *Kayal et al. (2018)* | 95 | 14,370 | SRX288276 |
| | *Physalia physalis* | *Kayal et al. (2018)* | 105 | 15,455 | SRX288431 |
| Aplanulata | *Hydra magnipapillata* 1 | *Kayal et al. (2018)* | 94 | 14,171 | SRX000112 |
| | *Hydra magnipapillata* 2 | *Chapman et al. (2010)* | 105 | 16,187 | PRJNA12876 |
| | *Hydra oligactis* | *Kayal et al. (2018)* | 64 | 8,342 | SRR040466–9 |
| | *Hydra viridissima* | *Kayal et al. (2018)* | 70 | 9,237 | SRR040470–3 |
| | *Ectopleura larynx* | *Kayal et al. (2018)* | 106 | 16,688 | SRX315375 |
| | | Outgroup | | | |
| Trachylina | *Craspedacusta sowerbii* | *Kayal et al. (2018)* | 116 | 18,413 | SRR923472 |
| | *Aglaura hemistoma* | USNM1622142 | 8 | 925 | MW271974–81 |
| | *Solmundaegina nematophora* | USNM1284330 | 6 | 324 | MW271968–73 |
| | *Aegina citrea* | *Kayal et al. (2018)* | 69 | 9,713 | SAMN03418514 |
| Cubozoa | *Alatina alata* | *Kayal et al. (2018)* | 113 | 17,918 | SAMN03418513 |
| | *Tripedalia cystophora* | *Kayal et al. (2018)* | 34 | 4,088 | SRR1182852, SRR2103559 |
| Scyphozoa | *Periphylla periphylla* | *Kayal et al. (2018)* | 55 | 6,980 | SRR1915828 |
| | *Atolla vanhoeffeni* | *Kayal et al. (2018)* | 85 | 12,468 | SAMN03418515 |
| | *Chrysaora chesapeakei* | USNM1454941 | 37 | 3,145 | MW272392–428 |
| | *Cyanea capillata* | *Kayal et al. (2018)* | 37 | 4,503 | SRR1930118 |

**Table 2** (*continued*)

|  | Taxon | Voucher/Reference | Loci | Residues | NCBI Accession |
|---|---|---|---|---|---|
|  | *Stomolophus meleagris* | *Kayal et al. (2018)* | 103 | 15,551 | SRR1168418 |
|  | *Cassiopea xamachana* | *Kayal et al. (2018)* | 93 | 12,888 | ERX2270394–7, ERX2281610–3 |
|  | *Aurelia aurita* | *Brekhman et al. (2015)* | 111 | 17,342 | GBRG00000000 |
| Staurozoa | ***Haliclystus auricula*** | **USNM1621043** | **17** | **1,406** | **MW271951–67** |
|  | *Craterolophus convolvulus* | *Kayal et al. (2018)* | 94 | 12,821 | ERR2248381 |
|  | *Calvadosia cruxmelitensis* | *Kayal et al. (2018)* | 109 | 16,800 | SRR13003944 |

cnidarian reference protein collection from *Kayal et al. (2018)* using translated BLAST queries. After extraction of target reads and binning by locus, reads were assembled for each species using the SPAdes assembler (version 3.10.1; *Bankevich et al., 2012*), followed by alignment and scaffolding of contigs against the reference proteins using Exonerate (version 2.2.0; *Slater & Birney, 2005*). In-frame coding sequences of scaffolds (super-contigs) were translated into amino acids. All loci were individually aligned using MAFFT (version 7.271; *Katoh & Standley, 2013*) and ambiguous alignment positions were removed using Gblocks (version 0.91b; *Talavera & Castresana, 2007*) *via* Gblockwrapper (version 0.03; https://goo.gl/fDjan6). Gene trees and their support from 1,000 non-parametric bootstrap replicates were inferred using RAxML (version 8.2.12; *Stamatakis, 2006*) under the best fit model of sequence evolution (LG, WAG, or MtZoa; see phylogenetic inference below) using the computational resources provided by the Open Science Grid (*Pordes et al., 2007*; *Sfiligoi et al., 2009*). Gene trees were summarized by maximizing the number of shared quartet trees using ASTRAL-III (version 5.7.7; *Zhang et al., 2018*). Using the resulting phylogenetic tree and quartet frequenceies enabled us to identify areas of gene tree discordance. Following alignment and removal of ambiguous alignment positions, all loci were concatenated for further phylogenetic analyses.

## Phylogenetic inference

Phylobayes (version 4.1c; *Lartillot et al., 2013*) was used to run eight independent MCMC chains and the posterior probability distribution was sampled until chains converged and a large sample of trees was generated. To account for site-specific differences in the evolutionary rates within and among alignment partitions, site-specific rates were inferred from the data using the CAT-GTR model during MCMC runs (*Lartillot & Philippe, 2004*). PartitionFinder (version 2.1.1; *Lanfear et al., 2017*) was used to determine the best partitioning scheme for the concatenated alignment, with maximum likelihood trees for the partitioning analysis inferred using RAxML (version 8.2.12; *Stamatakis, 2006*). In the absence of prior information on possible partitioning schemes, the relaxed clustering algorithm (rclusterf; *Lanfear et al., 2014*) was used to identify partitioning schemes that fit the data well. Initial clustering analyses included all substitution models implemented in PartitionFinder. These exploratory analyses failed to finish after more than a month of run-time but indicated that models with rate heterogeneity modeled by drawing from the gamma distribution (+G) and amino acid residue equilibrium frequencies estimated from the data (+F) fit data partitions best. Due to their prevalence in preliminary results and tractability of partitioning analysis, the final partitioning scheme was inferred using the LG

(*Le & Gascuel, 2008*), WAG (*Whelan & Goldman, 2001*), and MtZoa (*Rota-Stabelli, Yang & Telford, 2009*) substitution matrices. The resulting partitioning scheme was used to infer the maximum likelihood phylogeny using RAxML (*Stamatakis, 2006*). The best tree was chosen from a set of 10 trees inferred from independent searches, starting from different random starting trees. Robustness of the resulting maximum likelihood phylogeny was assessed using 681 non-parametric bootstrap replicates.

## Tree topology hypothesis testing

Minimally constrained maximum likelihood phylogenetic tree searches were conducted to evaluate which lineage of Hydroidolina is the sister group to the remainder of Hydroidolina: Aplanulata, Capitata (*Cartwright & Nawrocki, 2010*), or Siphonophorae (*Kayal et al., 2015*). In addition, tree topologies were fully constrained following the phylogenetic hypotheses proposed in this contribution and previous phylogenetic hypotheses of Hydroidolina (*Cartwright & Nawrocki, 2010*; *Kayal et al., 2015*). To discriminate between competing hypotheses of hydroidolinan relationships inferred under these comprehensive constraints, phylogenetic analyses were conducted using maximum likelihood followed by topological hypothesis tests. In short, the backbone of each tree, including the best trees found in unconstrained searches presented in this contribution, was fixed at nodes that define higher-level relationships of in- and outgroups. Under these constraints, tree inferences were able to rearrange the topology of unconstrained nodes and make adjustments to branch lengths to maximize the likelihood of the tree given the data in the concatenated amino acid residue alignment. These constrained maximum likelihood phylogenies were reconstructed using 10 independent partitioned RAxML searches, retaining the tree with the highest log likelihood (lnL). Further, the two competing hypotheses presented in this contribution were compared using fully constrained Bayesian inferences. By using fully constrained starting phylogenies, we set informative priors on tree searches *sensu Bergsten, Nilsson & Ronquist (2013)*. Samples of the Bayesian posterior probability distribution were obtained from four independent MCMC chains under both the CAT-GTR and default CAT-Poisson (F81) models implemented in PhyloBayes, as described under phylogenetic inference above. Sampling the posterior of tree searches under two different models allowed us to identify the sensitivity of the phylogenetic inference to model misspecifications.

The Bayesian inference presented herein was used as the null hypothesis ($T_0$) against which all alternate trees were evaluated quantitatively using the likelihood ratio. The likelihood ratio statistic was calculated as

$$\delta lnL = 2(lnL\ T_A - lnL\ T_0)$$

where lnL $T_0$ represents the likelihood of the tree under the null hypothesis and lnL $T_A$ the likelihood of the tree under the alternate hypothesis. Here, a positive $\delta lnL$ indicates a better fit of $T_0$ to the data while a negative $\delta lnL$ indicates that $T_A$ explains the data better. Resampling of site log likelihoods (RELL; *Kishino, Miyata & Hasegawa, 1990*; *Hasegawa & Kishino, 1994*) was used to generate 10,000 bootstrap samples for each tree hypothesis, estimating the variance of tree likelihoods. Using these RELL bootstrap distributions, the null hypothesis that trees of the candidate set have the same lnL was tested *via* the

approximately unbiased test (AU; *Shimodaira, 2002*), as implemented in Consel (version 0.2; *Shimodaira & Hasegawa, 2001*). Phylogenetic tree fit to the data was further evaluated using the posterior probability of all candidate trees using the Bayesian Information Criterion (BIC; *Schwarz, 1978*) approximation implemented in Consel.

Bayes factors (reviewed in *Morey, Romeijn & Rouder, 2016*) were calculated to compare the Bayesian majority rule consesus topology ($T_0$) with the alternate maximum likelihood topology ($T_1$) inferred in this constribution under the CAT-GTR and CAT-Poisson (F81) models in pairwise comparisons. These comparisons address the question of how well the best tree hypothesis predicts the observed amino acid alignment compared to the alternate tree hypotheses under a given substitution model. Bayesian model evaluation requires quantifying model evidence by a marginal likelihood function through integration of the product of the likelihood and the prior (*Fourment et al., 2020*). Model comparisons with Bayes factors use ratios of marginalized likelihoods, which are difficult to compute exactly. To address this issue, the harmonic mean of the posterior is widely used as an estimator of the marginal likelihood (*Kass & Raftery, 1995*). Combining the posterior distributions of four MCMC chains for a given constrained tree search, we calculated the moving harmonic mean of the posterior using a sliding window with a step length of one and a size equal to the current MCMC cycle times 0.33. The natural logarithm of the ratio of harmonic means was used for tree comparisons and interpreted following a modified version of Jeffreys' (*Jeffreys, 1961*) categories of evidence (cf. *Kass & Raftery, 1995*). In particular, Bayes factors were calculated as

$$BF_{10} = 2 * ln(lnL\ T_1 - lnL\ T_0)$$

where lnL $T_0$ is the marginal likelihood of the MCMC search under the null hypothesis and $T_1$ the marginal likelihood of the alternate MCMC tree search. Note that the likelihood ratio may be negative leading to an undefined result of the logarithmic function, an issue we took into account by taking the absolute of the ratio if negative, followed by multiplying the resulting Bayes factor by negative one.

## RESULTS

### Recovery of target loci and alignment

The final concatenated alignment (Fig. 1A; Supplemental File 2) contained 44 medusozoan cnidarians, sequence data for 18 of which were generated in this study (Table 2). This alignment contained 134 of the 355 targeted loci, with a total alignment length of 21,816 character columns. Loci targeted but not included in the final alignment failed to generate sequence data from target enrichment reactions and were excluded from further analysis. Alignment lengths per locus ranged from 56 to 363 positions (average 164), with a combined total of 333 to 5,818 (average 2,891) amino acids contained in each alignment (Fig. 1B). Taxon occupancy per locus ranged from 5 to 36 species with an average of 21 species included in each alignment partition for phylogenetic analysis (Fig. 1C). Among ingroup hydrozoans, coverage varied between and within clades, with matrix completeness being highest for taxa of the *Kayal et al. (2018)* reference dataset (Fig. 1A). In particular, sequence

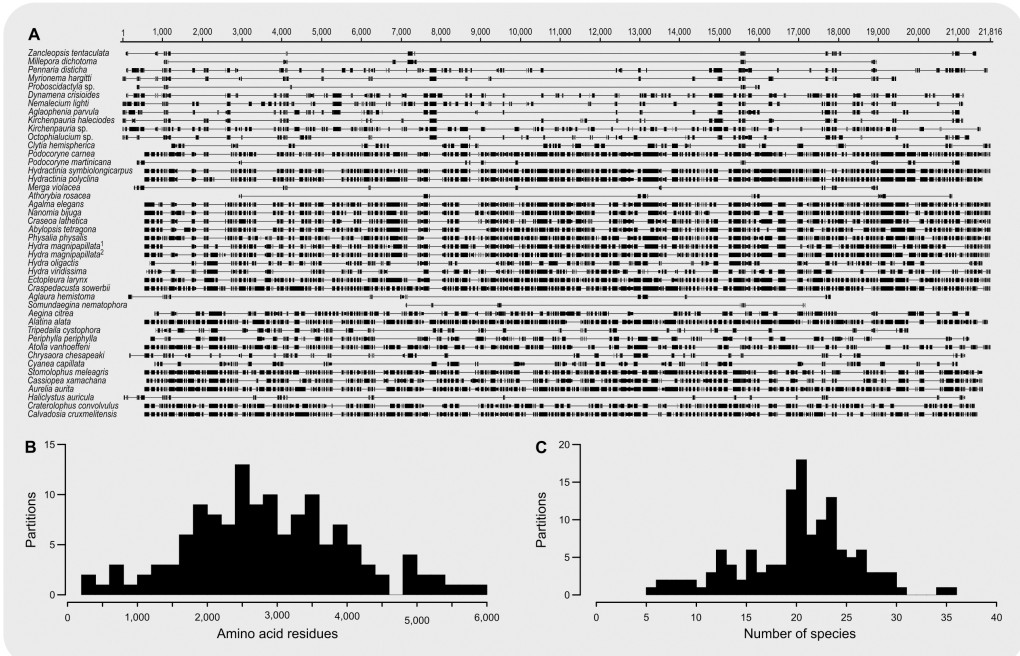

**Figure 1** **Summary of amino acid alignment completeness and taxon coverage.** (A) Alignment of the 134 amino acid residue partitions; (B) total number of amino acid residues contained in each alignment partition; (C) number of species included in each alignment partition.

data for Aplanulata, Siphonophorae, and Filifera III largely represent previously published sequence data that were used for bait design (Table 2). Consequently, alignment coverage in these groups ranges from some 40% to almost 80%. Despite being closely related to at least some of the taxa used in bait design, success in recovering loci from *Podocoryne martinicana* (Filifera III) and *Athorybia rosacea* (Siphonophorae) was limited (Table 2). In species of Filifera I, Filifera II, and Capitata target capture yields ranged from as few as some 500 amino acid residues spread over five loci (*Myrionema hargitti*) to as high as some 4,400 residues contained in 46 separate loci (*Pennaria disticha*). By contrast, the RNA bait set and hybridzation protocol employed here was highly successful in recovering sequence data for leptothecate hydrozoans despite being somewhat distantly related to any of the reference taxa. On average, 49 target loci comprising close to 5,000 amino acid residues were recovered for leptothecate species (Table 2). Here, the maximum yield was some 8,000 residues spread across 80 separate loci (*Kirchenpaueria* sp.). Consistent with the high degree of missing sequence data, gene trees displayed a large amount of discordance for both ingroup and outgroup taxa (Fig. 2).

## Phylogeny of Hydroidolina inferred from target-capture sequencing
Bayesian MCMC chains using the CAT-GTR model were terminated after some 14,000 iterations, of which the first 3,000 were discarded as burn-in. Chains were thinned by sampling every tenth generation, yielding a mean difference between chains of 9e−4 and a maximum difference of 0.0469. The majority rule consensus of the posterior shows

Peer J

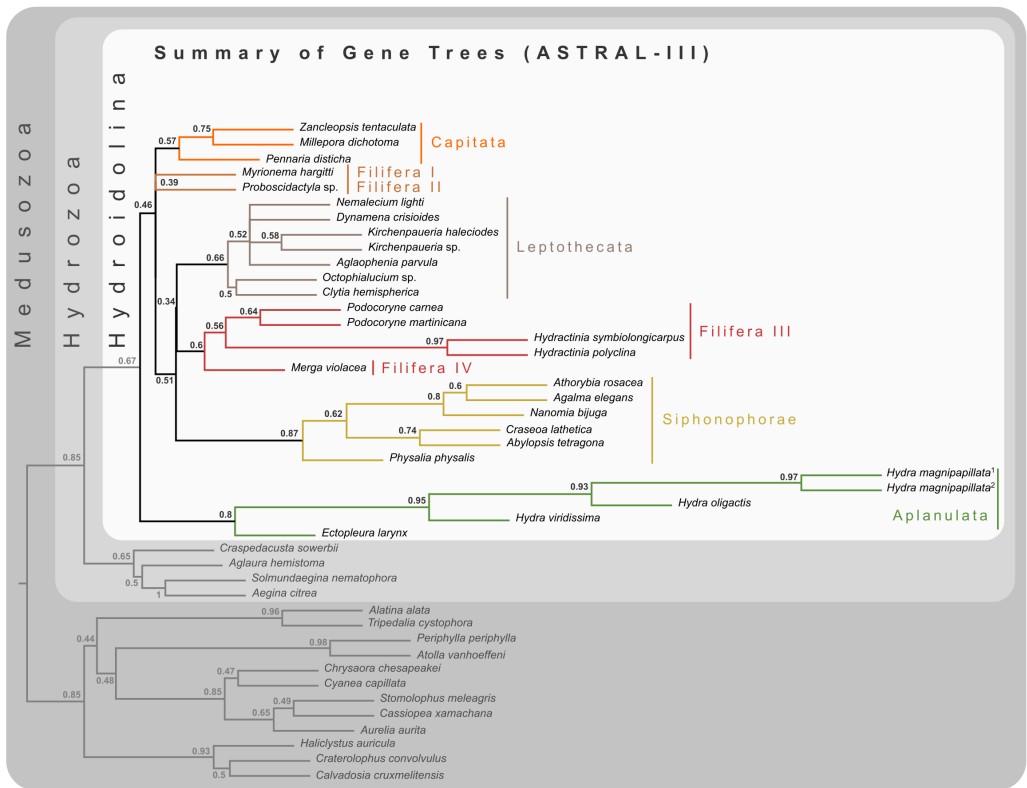

**Figure 2 Cladogram of statistical gene tree summary derived from the most common quartet trees obtained from 134 bootstrapped gene trees.** Node labels represent mean quartet frequencies and give insight into areas of gene tree conflict in the dataset. Quartet frequencies of 1 indicate no gene tree discordance while lower frequencies indicate discordance (*Mirabab, 2019*).

a well-resolved phylogeny with high posterior probabilities for the monophyly of the major ingroup and outgroup clades (Fig. 3). Aplanulata represents the sister group to the remainder of Hydroidolina. As in previous analyses, Filifera represents a polyphyletic taxon. Interestingly, Filifera I plus II are each other's closest relative as are Filifera III plus IV. Filifera I plus II are the closest relatives to Capitata, albeit with a posterior probability slightly less than 0.95; Leptothecata is inferred as sister to the clade of Filifera I plus II and Capiata. Filifera III plus IV are the closest relatives of Siphonophorae.

For ML phylogenetic inference, the best partitioning scheme of the 134 locus dataset contained 25 partitions. In particular, the ML phylogeny was inferred using 22 partitions comprising 128 loci under the LG+G+F model, two partitions with a total of 5 loci under the MTZOA+G+F model, and 1 partition with a single locus under the WAG+G+F model. The ML tree largely agrees with the Bayesian inference (Fig. 3), but differs in the placement of Filifera I plus II relative to Capitata and Leptothecata. In contrast to the Bayesian inference, Filifera I plus II are the sister to Leptothecata rather than Capitata. However, this result lacks support with a bootstrap of less than 50. While the majority of the other nodes in both the in- and outgroup received high bootstrap support, the deep ingroup node uniting Filifera III plus IV and Siphonophorae is weakly supported with a bootstrap

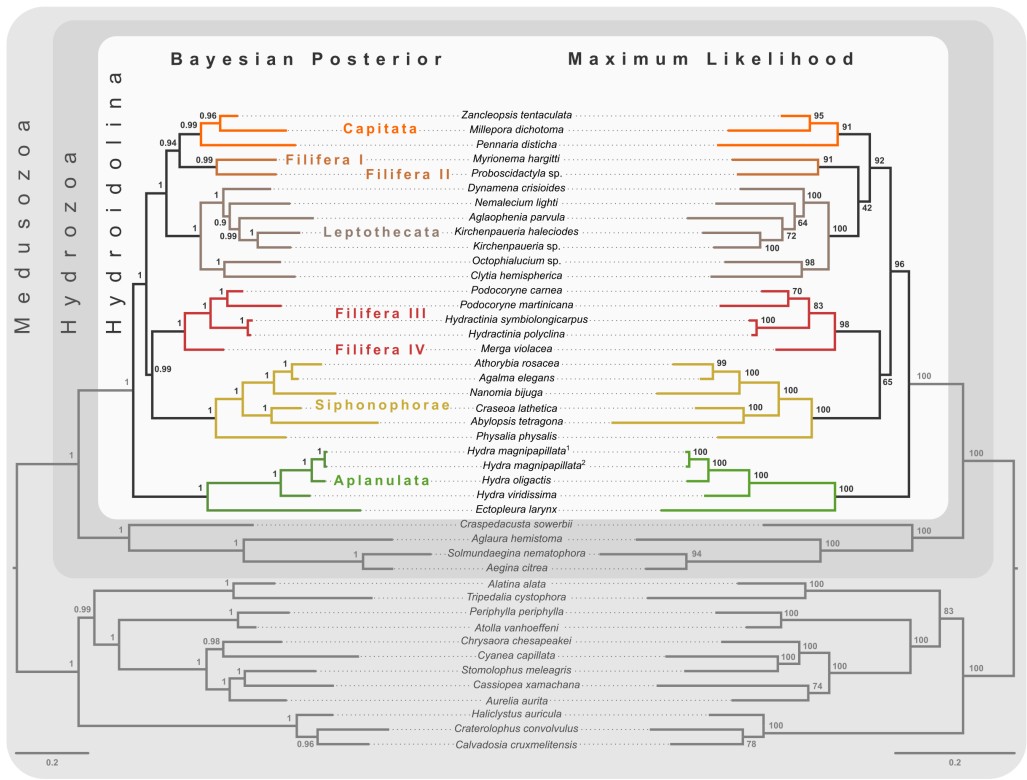

**Figure 3 Bayesian and maximum likelihood phylogenies inferred from concatenated amino acid alignment.** Majority rule consensus of Bayesian phylogenetic inference under the CAT-GTR model (left side) with posterior probabilities given at each node. Maximum likelihood phylogeny (right side) was inferred from a partitioned analysis under the LG, WAG, and MtZoa models. Node support values represent the summary of 681 non-parametric maximum likelihood bootstrap searches. Scale bars indicate substitutions per site on each respective phylogram.

of 65. Despite these differences between Bayesian and ML phylogenies, both analyses agree in confidently inferring Aplanulata to be the sister lineage of the rest of Hydroidolina.

## Phylogenetic hypothesis testing

Aplanulata represents the most likely sister taxon to the remainder of Hydroidolina (Fig. 4 left) with the alternate hypotheses of Capitata (Fig. 4 center) or Siphonophorae (Fig. 4 right) both receiving low BIC posterior probabilities and being rejected by the AU test. Fully constrained phylogenetic inferences provided further insight into the relationships within Hydroidolina (Fig. 5A). Two phylogenies, $T_2$ (cf. *Cartwright & Nawrocki, 2010*) and $T_3$ (cf. *Kayal et al., 2015*), were rejected by the AU test (Table 3). Interestingly, $T_2$ had a higher likelihood than $T_3$, indicating a worse fit of $T_3$ to the data compared to $T_2$. $T_0$ and $T_1$ were constrained following the Bayesian and Maximum likelihood phylogenies inferred herein (Fig. 3). Both tree hypotheses are roughly equally likely, with the maximum likelihood phylogeny $T_1$ being a slightly better fit to the data using maximum likelihood tree searches (Table 3). Constrained Bayesian searches using $T_0$ and $T_1$ as priors generated a sample of 27,000 log likelihoods from the posterior, with chains mixing well after burn-in

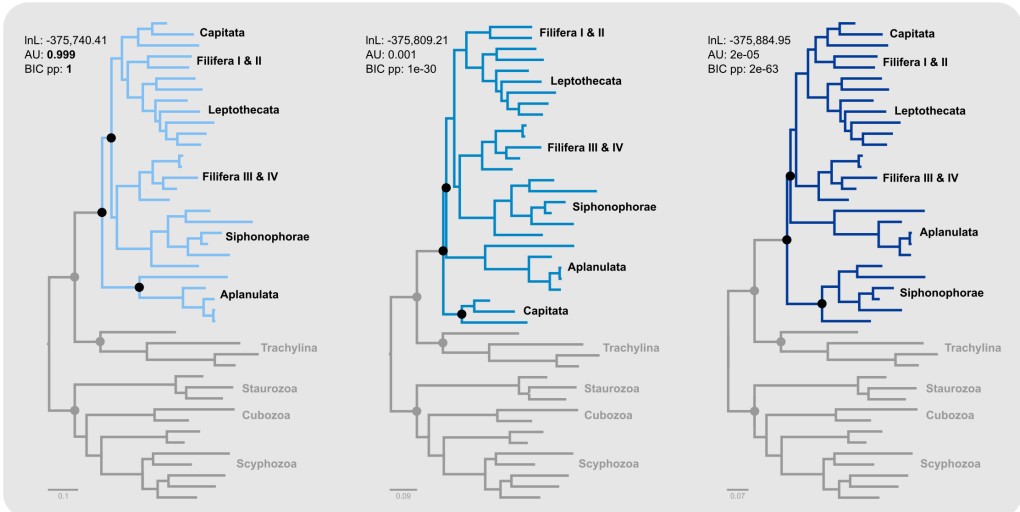

**Figure 4** **Minimally constrained maximum likelihood phylograms.** Constrained nodes are indicated by solid circles and the log likelihood for each phylogeny are given; scale bars indicate substitutions per site. The backbone of the phylogeny was constrained to maintain monophyly of the outgroup and monophyly of Hydroidolina with respect to Trachylina. Within Hydroidolina, three hypotheses of sister-group relationships were evaluated: Aplanulata (left), Capitata (center), or Siphonophorae (right) as sister to the remainder of Hydroidolina. The approximately unbiased test (AU) indicated that the Capitata (center) and Siphonophorae (right) sister-group hypotheses should be rejected. The Bayesian Information Criterion (BIC) showed a good fit of the Aplanulata (left) sister-group phylogeny to the alignment data.

(Fig. 5B). The marginal likelihood of $T_0$ was higher than the marginal likelihood of $T_1$ under the GTR model while the marginal likelihood of $T_1$ was higher under the F81 model (Fig. 5C). Under the GTR substitution model, $T_0$ is better at predicting the data than $T_1$ ($BF_{10} = 10.48$; Fig. 5D). By contrast, $T_1$ is favored under the F81 model ($BF_{10} = -8.84$; Fig. 5D).

## DISCUSSION

### A new phylogenetic hypothesis for Hydroidolina

Despite highly variable recovery rates of target loci that leave large gaps in the final alignment (Fig. 1A), the dataset analyzed here provides high resolution of the deep phylogeny of Hydroidolina (Fig. 3) even though discordance among gene trees is apparent (Fig. 2). Considering the ancient radiation of medusozoan cnidarians, the gene tree discordance observed here is unlikely a result of incomplete lineage sorting but rather the result of gene tree estimation error driven by the limited information contained in individual amino acid alignments. Coalescent-based summary methods lack accuracy in the presence of substantial gene tree estimation error (*Warnow, 2015*) and, under these conditions, phylogenetic analyses of fully partitioned concatenated alignments are preferable over summary methods that implement multi-species coalescent models (cf. *Molloy & Warnow, 2018*; *Bryant & Hahn, 2020*). While missing data could affect tree topology inference from concatenated alignments, in practice accurately placing taxa despite missing information

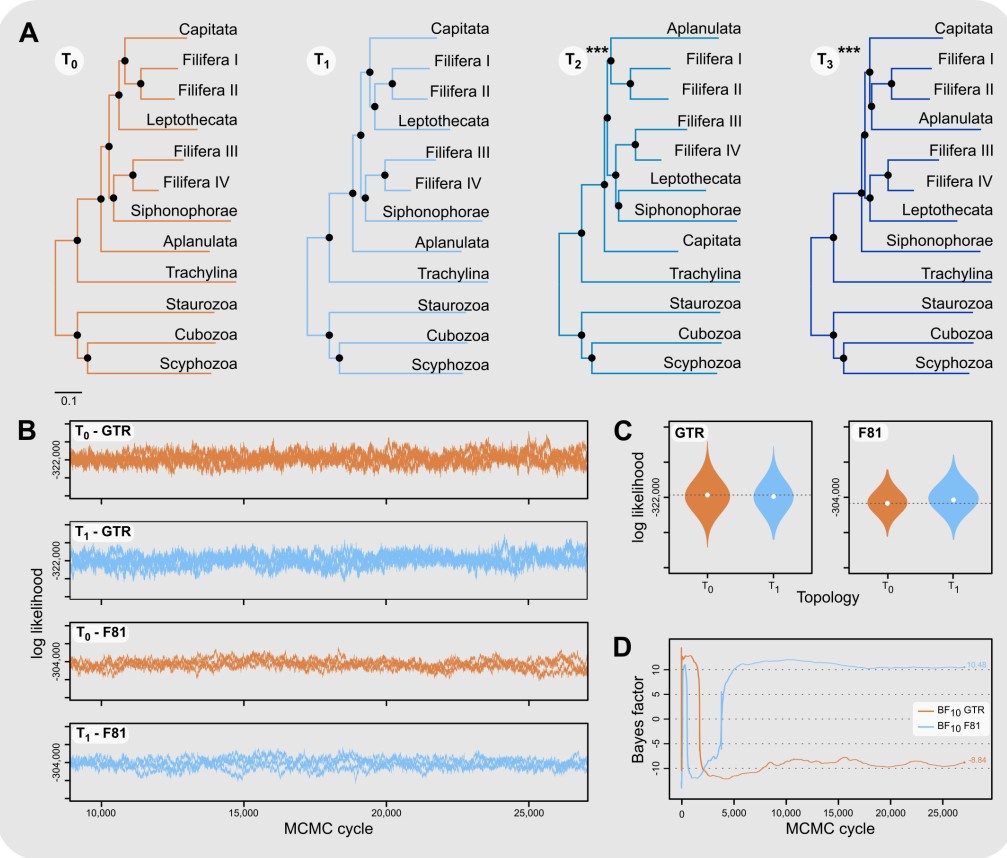

**Figure 5** **Phylogenetic hypothesis tests using fully constrained tree searches.** (A) Phylograms inferred using fully constrained maximum likelihood searches to compare explicit hypotheses of tree topologies; constraints indicated by solid black circles; scale bar indicates substitutions per site. Tree topologies were constrained to reflect the Bayesian inference ($T_0$) and maximum likelihood phylogenies ($T_1$) presented in this contribution (Fig. 3). Two alternate tree searches were constrained to reflect previously published phylogenies ($T_2$: *Cartwright & Nawrocki, 2010*; $T_3$: *Kayal et al., 2015*); *** indicates a *p*-value of less than 0.001 for the approximately unbiased (AU) test (Table 3). Shallow nodes were collapsed and species labels removed for legibility. (B) Each graph shows the log likelihoods of four independent Bayesian Markov Chain Monte Carlo (MCMC) runs following burn-in. Log likelihoods for both $T_0$ and $T_1$ were sampled from the posterior under the CAT-GTR and the CAT-Poisson (F81) models using the constraints shown in (A). (C) Harmonic means representing the marginal likelihood (white circles) and log likelihood distributions of $T_0$ and $T_1$ under CAT-GTR (left) and CAT-Poisson (F81; right) models. Grey lines indicate the harmonic means for $T_0$. (D) Bayes factors for pairwise comparisons between $T_0$ and $T_1$ (C). A negative Bayes factor provides evidence for $T_0$ while a positive Bayes factor indicates evidence against $T_0$, providing support for $T_1$. A Bayes factor between $-2$ and 2 indicates lack of evidence or no evidence favoring one hypothesis over the other, Bayes factors in the range of 2 to 6 ($-2$ to $-6$) provide some evidence, 6 to 10 ($-6$ to $-10$) strong evidence, and Bayes factors $> 10$ ($< -10$) very strong evidence for one hypothesis compared to the other.

is often not a major concern (*Wiens & Morrill, 2011*). Indeed, we were able to infer a well-resolved phylogeny by combining a publicly available data-rich amino acid sequence dataset with new data from key hydrozoan taxa that have so far been absent from multigene phylogenomic datasets. Overall topologies recovered in our phylogenetic inferences

**Table 3** Likelihood ratios were calculated to compare alternate tree hypotheses to the null, $T_0$ (cf. *Cartwright & Nawrocki, 2010*). The approximately unbiased test (AU) was used to test the null hypothesis that all tree topologies have the same log likelihood. Tree topology fit to the alignment data was evaluated using the posterior probability of the Bayesian Information Criterion (BIC).

| Constraint | Log likelihood | Likelihood ratio | AU | BIC posterior |
|---|---|---|---|---|
| $T_0$ | −375,741.33 | – | 0.54 | 0.282 |
| $T_1$ | −375,740.41 | −0.92 | 0.60 | 0.718 |
| $T_2$ | −375,844.52 | 103.19 | 2e−4 | 4e−46 |
| $T_3$ | −375,944.68 | 203.35 | 4e−4 | 1e−89 |

are consistent with previous phylogenomic analyses (*Zapata et al., 2015*; *Kayal et al., 2018*). While neither *Zapata et al. (2015)* nor *Kayal et al. (2018)* included sufficient taxa representing Hydroidolina to allow for much insight into the evolutionary history of the group, both recovered Aplanulata as the sister clade of the remaining hydroidolinan groups, a placement we confirmed after comprehensively sampling higher-level taxa of Hydroidolina (Fig. 3). This placement is at odds with previous phylogenetic treatments of Hydroidolina that inferred a more recent origin of Aplanulata within Hydroidolina as sister of Filifera I plus II (*Cartwright & Nawrocki, 2010*; *Kayal et al., 2015*). Placing Aplanulata as the sister-group of Hydroidolina has potentially far-reaching consequences for our understanding of hydrozoan evolution. Aplanulata is dominated by solitary polyp-forming taxa, a trait common in other medusozoans but rare across Hydroidolina in which colonial hydroids, many of which are polymorphic, displaying a reproductive division of labor, dominate (*Cartwright & Nawrocki, 2010*; *Cartwright, Travert & Sanders, 2020*). As such, the placement of Aplanulata as sister to the remainder of Hydroidolina suggests that the last common ancestor of Hydroidolina possessed a solitary polyp that produces medusae as part of a metagenetic life cycle in which an asexually reproducing polyp generation alternates with a sexually reproducing medusa generation (cf. *Bentlage et al., 2018*). Pending formal analyses using ancestral state reconstructions with greater taxon sampling, the evolution of coloniality and division of labor, including losses of metagenesis (lack of free-swimming medusae), are likely derived traits in Hydroidolina.

Ingroup relationships within Hydroidolina appear largely congruent across the Bayesian posterior tree set and the maximum likelihood phylogeny with the exception of the placement of Filifera I plus II with respect to Capitata and Leptothecata (Fig. 3). One possible explanation for this disagreement may be that the substitution matrix-based models used for maximum likelihood inference here are potentially less accurate in reflecting the substitution process that led to the observed data. Exploratory partitioning analyses with PartitionFinder suggested that substitutions in a significant number of partitions should be inferred using the GTR model. However, benchmarking of partitioning analyses under the GTR model indicated that computational time would be prohibitive to search across this large parameter space. To address this issue, partitioning schemes were inferred using simpler substitution matrices (*i.e.*, LG, WAG, and MtZoa). Regardless of this poorly supported topological difference in the placement of Filifera I plus II (Fig. 3),

the overall relationships recovered are consistent across analyses but at odds with previous phylogenetic hypotheses of Hydroidolina (cf. *Cartwright & Nawrocki, 2010*; *Kayal et al., 2015*). Despite conflicts between the phylogenetic hypotheses presented here and previous treatments of Hydroidolina, a consistent result that has emerged over the last decade is the polyphyly of Filifera. In agreement with prior phylogenetic hypotheses (*Cartwright & Nawrocki, 2010*; *Kayal et al., 2015*), we find that Filifera I plus II forms a monophyletic clade, as does Filifera III plus IV. As of now, these clades remain without clear definition based on morphological or life history characteristics but *Cartwright et al. (2008)* provide the most detailed discussion of possible synapomorphies of hydroidolinan clades to date.

## Phylogenetic tree selection

Our results appear unequivocal on the position of Aplanulata as sister to the remaining lineages of Hydroidolina (Figs. 2–4). To facilitate discriminating between competing hypotheses of hydroidolinan relationships, we quantified the evidence for the set of alternate phylogenetic hypotheses (Fig. 5A) in light of the multi-locus dataset analyzed here. Likelihood ratio statistics (Table 3) indicate that the phylogenies proposed in this contribution (Fig. 3; Fig. 5A $T_0$ and $T_1$) are a better fit than either the previously published rDNA (Fig. 5A $T_2$; *Cartwright & Nawrocki, 2010*) or mitochondrial genome-based (Fig. 5A $T_3$; *Kayal et al., 2015*) phylogenetic hypotheses. Indeed, $T_2$ and $T_3$ were rejected by the AU test while neither $T_0$ nor $T_1$ could be rejected (Table 3). In addition, the majority rule consensus of the Bayesian posterior ($T_0$) and the maximum likelihood topology ($T_1$) received high BIC-based posterior probabilities compared to the very small posterior probabilities of the alternate hypotheses (Table 3).

The hypothesis tests above reduced the set of credible trees by eliminating unlikely tree topologies. The procedures employed rely on the likelihood of a single phylogeny, implicitly assuming that the tree is known without error. Resampling of site likelihoods from this best tree was used to estimate variances of tree likelihoods for comparisons. Bayesian approaches that estimate the posterior probability distribution of phylogenetic hypotheses allow averaging across tree topologies and branch lengths for more comprehensive incorporation of uncertainty into comparisons of phylogenetic trees but may be affected adversely by misspecifications of the prior (*e.g.*, *Bergsten, Nilsson & Ronquist, 2013*). Bayes factors incorporate phylogenetic uncertainty in model comparisons by employing ratios of marginal likelihoods and allowed us to quantify the weight of evidence for a topological hypothesis under a given amino acid substitution model. We found that $T_0$ predicts the alignment data best under the GTR model while $T_1$ predicts the alignment best under the simpler F81 model (Figs. 5C & 5D). The difference between these two topologies is limited to the placement of Filifera I plus II with respect to Capitata and Leptothecata (Fig. 3) and we suspect that the sensitivity to model choice may be driven by the lack of data for Filifera I plus II (Fig. 1) and the resulting gene tree discordance (Fig. 2).

Despite being unable to confidently choose a single phylogenetic hypothesis as fitting the data best using maximum likelihood and Bayesian hypothesis testing frameworks, the phylogenetic hypotheses presented in this contribution (Fig. 3) consistently display high predictive power for the alignment data compared to the alternate hypotheses proposed

previously. Consequently, we suggest that our phylogenetic framework for Hydroidolina represents the most viable hypothesis of hydroidolinan relationships to date. That said, further studies with additional taxa and characters are still needed to assess the validity of this working hypothesis and clarify the affinities of Filifera I plus II in particular. Such analyses will also allow for more thorough analyses of the evolution of life history characteristics across this most speciose of medusozoan clades.

## CONCLUSIONS

This study represents one of only a few attempts at using target-enriched high throughput sequencing to generate a multi-locus alignment for phylogenetic analyses of Cnidaria. Previous attempts had been limited to Anthozoa (*Quattrini et al., 2018*; *Erickson et al., 2021*), relying on the greater availability of genomic data for Anthozoa compared to Medusozoa. Despite mixed success in capturing target loci, we were able to generate an informative multi-gene alignment that produced a well-supported phylogenetic hypothesis for Hydroidolina. One drawback of bait-development based on coding sequences alone, as done here, is that baits may inadvertently be designed across intron-exon boundaries which is likely to reduce bait effectiveness. As availability of genomic resources increases, it will be possible to address these issues and apply target capture sequencing approaches routinely to phylogenetic studies of medusozoan cnidarians, building on the growing knowledge-base for designing targeted high-throughput sequencing experiments (reviewed in *Andermann et al., 2020*). Nonetheless, we were able to address long-standing questions in hydrozoan phylogenetics by applying relevant constraints for discriminating between alternate phylogenetic hypotheses for Hydroidolina. We suggest that the consensus of $T_0$ and $T_1$ (Fig. 5A) represents the most likely tree topology for hydroidolinan evolutionary relationships to date. While the likelihood-based AU test (Table 3) allowed rejection of several alternate topologies, Bayes factors (Fig. 5D) suggested that the choice of substitution model affects the posterior distributions of the two alternate topologies inferred. As such, the most viable representation of phylogenetic relationships of Hydroidolina is the consensus of the two topologies inferred herein.

## ACKNOWLEDGEMENTS

Part of this work was performed using resources of the Laboratories of Analytical Biology (LAB) at National Museum of Natural History (NMNH) and we thank LAB staff for valuable advice on laboratory protocols and procedures. The Open Science Grid (NSF award 2030508) and its support staff of research computing facilitators enabled high throughput computing for phylogenetic inferences. BB wishes to acknowledge the numerous discussions with the participants in his bioinformatics and data analysis course that informed many of the analytical approaches for phylogenetic hypothesis testing used here. We thank Casey Dunn and an anonymous reviewer for suggestions that improved this work.

### Funding

This work was supported by Guam EPSCoR through NSF awards OIA-1457769 and OIA-1946352, and funding by the Small Grants program of the NMNH, Smithsonian Institution. The funders had no role in study design, data collection and analysis, decision to publish, or preparation of the manuscript.

### Grant Disclosures

The following grant information was disclosed by the authors:
NSF Awards: OIA-1457769, OIA-1946352.
Small Grants program of the National Museum of Natural History (NMNH).
Smithsonian Institution.

### Competing Interests

The authors declare there are no competing interests.

### Author Contributions

- Bastian Bentlage conceived and designed the experiments, performed the experiments, analyzed the data, prepared figures and/or tables, authored or reviewed drafts of the paper, and approved the final draft.
- Allen G. Collins conceived and designed the experiments, authored or reviewed drafts of the paper, and approved the final draft.

### Data Availability

   All sequences are available in NCBI GenBank (Table 2): MW271951–MW272434

### Supplemental Information

Supplemental information for this article can be found online at http://dx.doi.org/10.7717/peerj.12104#supplemental-information.

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
