# Peer review of "Tackling the phylogenetic conundrum of Hydroidolina (Cnidaria: Medusozoa: Hydrozoa) by assessing competing tree topologies with targeted high-throughput sequencing"

_PeerJ, doi:10.7717/peerj.12104_

## Round 0.1 · original submission · Minor Revisions

The two reviewers both agree that this study is exciting, well-written, and contributes to our understanding of the cnidarian tree of life. They both, however, suggest several methodological improvements, including using a multispecies coalescent model for the phylogenetic inferences. Please address these analytical concerns accordingly in your revision.

Reviewer 1 ·

Basic reporting

The manuscript is well written and concise in its current form but the introduction could include more detail about cnidarian evolution and hydrozoan biology. At the moment, this is very briefly discussed in the first paragraph, with specific reference to hydrozoan traits on lines 39-40, but it could be expanded and placed in the context of cnidarian evolution more broadly. For instance, did hydrozoans diversify in a similar way (i.e. through bursts of diversification?) as other cnidarian classes? How do their life cycle strategies compare with other groups? I understand this is more of a methodological paper but I think it's important to frame this properly.
Figure 2 could be improved by colour coding branches within Hydroidolina rather than using more grey boxes to define each taxonomic group. This would help focus the reader's eye on the group of interest.
In the interest of data accessibility and transparency, it would be helpful to have the NCBI accessions included in Table 1. Accessions should also be provided in Table 2 for previously published data. It would be best to provide the subclass (or outgroup) label horizontally at the top of the first column rather than vertically.

Experimental design

The authors have significantly improved understanding of hydrozoan evolution and phylogeny by including many taxa that were omitted in previous studies. The laboratory and bioinformatic approaches are certainly sound and described with sufficient detail. The results are drawn from two methods of phylogenetic inference that largely agree with one another but both rely solely on concatenated data. We know that data concatenation can be problematic in the face of gene tree discordance, which isn't limited to just shallow relationships (i.e. deep coalescence and ancient introgression). In light of this, the authors should consider implementing a multispecies coalescent model for phylogenetic inference, even just a summary coalescent approach like ASTRAL. The paper by Liu et al. (2019) (https://www.nature.com/articles/s41467-019-09454-w#Sec1) uses a similar approach for resolving the phylogeny of mosses, a diverse group that includes 30 different orders.

Validity of the findings

As stated above, I would encourage the authors to implement a multispecies coalescent model for phylogenetic inference to compliment their current analyses and support the framework provided here.
On lines 345-346 the authors state that the Filifera clades remain without a clear definition. Does this simply reflect the fact that Filifera is a particularly understudied group within the subclass? Or have attempts been made to examine the morphology and anatomy of these groups and still no defining characters have been discovered? It might be worth clarifying this here.
Otherwise, the conclusions are well stated and linked to the author's original objectives.

·

Basic reporting

Lines 24-25: "The Bayesian posterior topology infers Aplanulata as earliest branching lineage of Hydroidolina" I suggest revising this to "The Bayesian posterior topology infers Aplanulata as the sister group to all other members of Hydroidolina". Neither of these two clades is earlier branching than the other. This applies to all occurrences of "earliest diverging" throughout the manuscript.

Line 54: Comma after parenthesis should be a period.

Line 182, 283: It isn't clear how or why the topologies that are examined in these section were selected. They aren't related to specific results in previous studies.

Line 197: Why weren't these topologies all compared to the results of the present study instead? It has much better sampling and is an exciting advance over these previous analyses.

Line 323: "far-reaching consequences" be more specific here. Especially after featuring it in the abstract, was surprised to not have more discussion of this. I think it is very interesting, but am not exactly sure what consequences the authors have in mind. It reads as if implying that the MRCA of Hydroidolina was solitary, but this is not clear. It could be useful to include a tree where these traits are explicitly mapped.

Lines 374-376: This is quite speculative and I think only captures part of the issue. Mitochondrial genomes are really small - there just aren't many data to work with. Rate can play a role too, but it is clear that many difficult challenges in cnidarian phylogeny require considerable more data than are available in mt genomes.

Line 403: Remove or reword. The maximum likelihood tree is the literally the most likely tree.

Experimental design

The only comment I have regarding experimental design is the topological investigation summarized in Figure 3. My major concern is that so many nodes are constrained in each topology is very difficult to understand what features of these topologies are driving the differences in likelihood. It is much more informative to constrain only one or maybe two nodes at a time to really understand the impact of topology on likelihood.

Rather than constrain many nodes to compare the entire topologies, I suggest identifying a list of specific relationships that have been variable across analyses, and then constraining those one by one.

Validity of the findings

This is a very sound body of work, and I have no concerns about validity apart from the issues noted above in experimental design.

Additional comments

This is a very interesting and exciting paper. It will be of great interest given the fascinating diversity of traits in these groups. It makes important advances on key challenges that have been highlighted over the last decade as many other relationships in the group firmed up.

---

## Round 0.2 · Minor Revisions

The revised manuscript addressed most of the reviewers' previous concerns. Reviewer 2 pointed out two minor issues. Please revise the figures accordingly before the paper can be accepted.

Reviewer 1 ·

Basic reporting

The authors have addressed my comment about hydrozoan biology by adding relevant text to the Discussion. The revised version of the manuscript is well written and makes a significant contribution to hydrozoan evolution and phylogeny.

Experimental design

The authors addressed my concern about data concatenation and have revised accordingly. The methods here are sound, described with sufficient detail and address the objectives set out by the authors.

Validity of the findings

The authors addressed my concern about data concatenation and have revised accordingly. Additionally, all underlying data have been provided.

Additional comments

This is a great piece of work and I look forward to seeing it published!

·

Basic reporting

Be sure that each phylogeny figure identifies whether the tree is a cladogram or phylogram, and if it is a phylogram that there is an indication of meaning of branch length.

On figure 4, I believe the constrained nodes are shown incorrectly. In all cases, the solid circle indicates Hydroidolina, which has the same composition in each analysis and so does not differ. The constraint should be on the in-group that excludes the tested sister. From left to right, that is the clade Capitata...Siphonophora, Filifera...Aplanulata, Capitata...Siphonophora.

Experimental design

No issues noted.

Validity of the findings

No issues noted.

Additional comments

The authors addressed all my previous concerns, and this is going to be a valuable contribution to the community.

---

## Round 0.3 · accepted · Accept

Thanks for making the requested adjustments on the figures. The paper is now in good condition to be published.